# Challenges of Promoting Open Science within the NI4OS-Europe Project in Hungary

Ákos Lencsés [1,2,*] and Péter Sütő [3,*]

1   Governmental Agency for IT Development, 1134 Budapest, Hungary
2   Department of Cultural Studies, Kodolányi János University, 1139 Budapest, Hungary
3   University Library, University of Public Service, 1083 Budapest, Hungary
*   Correspondence: lencses.akos@kifu.gov.hu (Á.L.); suto.peter@uni-nke.hu (P.S.)

**Abstract:** National Initiatives for Open Science in Europe (NI4OS-Europe) is a Horizon 2020 project related to the European Open Science Cloud (EOSC). One of the project objectives is promoting EOSC and open science in 15 Central and East European EU states and EU-associated countries. This paper describes the variety of promoting activities carried out in Hungary as part of the NI4OS-Europe project by the Governmental Agency for IT Development (KIFÜ). Identifying good practices will give us the chance to find the best communication channels and methods to promote open science and to manage expectations of funders, researchers and librarians. The audience diversity of organized NI4OS events was analyzed in this study. The anonymized dataset based on registration forms was filtered by profession. Results suggest that events are generally visited by more librarians than researchers. The only exception is the third forum where the main Hungarian research fund as co-organizer might have attracted researchers' attention. This suggests that librarians are considered to be in charge of open science issues in general. Usage data of the open science news feed were also studied. The 130 posts between May 2021 and April 2022 and 2500 visitors until the end of June 2022 give us the chance to learn about the characteristics of the most visited posts. We can conclude that the focus of communication is on open and FAIR data management, while other areas receive less attention. The results show that despite more international posts being published, the target group is more interested in local information.

**Keywords:** Hungary; NI4OS-Europe; EOSC; open science; science communication

## 1. Introduction

National Initiatives for Open Science in Europe (NI4OS-Europe) is a Horizon 2020 project related to the European Open Science Cloud (EOSC) that runs between 1 September 2019 and 28 February 2023. One of the project objectives is promoting EOSC and open science in 15 Central and East European EU states and EU-associated countries. In the case of Hungary, two actively cooperating institutions take part in the consortium. The University of Debrecen University and National Library (DEENK) is the central library of one of the largest higher education institutions of the country. With a history reaching back to 500 years, 14 faculties and 30,000 FTE, University of Debrecen is among the top universities in Hungary. The other consortium member, the Governmental Agency for IT Development (KIFÜ) is the Hungarian national research and education network (NREN) provider. KIFÜ serves digitalization in Hungary, having 6400 customers and 2.5 million users; it offers a wide range of IT services for research and higher education.

This paper describes the variety of promoting activities carried out in Hungary as part of the NI4OS-Europe project by the Governmental Agency for IT Development (KIFÜ) in 2021 and H1 2022. An overview of these activities and identifying good practices will give us the chance to find the best communication channels and methods to promote open science and to manage expectations of funders, researchers and librarians in Hungary.

## 2. Materials and Methods

EOSC, starting its operation after many years of discussions in 2018, was originally aimed at managing European research data via European infrastructure [1]. Today, EOSC promotes not only FAIR principles but all other aspects of Open Science and has become a major player of the new, open research paradigm in Europe.

Several papers already highlighted the gap between the consensus of the importance of open research culture and the daily routine of researchers [2–5]. However, a relatively small amount of literature is available on best practices in promoting open science among researchers or analyzing best drivers that could help researchers embrace open science.

A detailed paper has been published recently by Robson et al. [6] that discusses mostly the psychological aspects of promoting open science. This publication analyzes the needs of different target groups (e.g., stakeholder groups, individual researchers) and the approaches that help us to understand what various practices are needed to reach out to these groups. In this paper, we also attempt to group NI4OS-Europe promoting activities by target groups.

Examining the conditions in Hungary, the engagement to open science goes back to the Budapest Open Access Initiative in 2001. In the next 1.5 decades, development of green open access was given priority, which led to the appearance of about 40 institutional repositories, some institutional OA policies and the creation of the Hungarian Open Repositories consortium (HUNOR). During this period, the open science communication focused on promoting green open access and on creating the related infrastructures and strategies [7–9]. Unsurprisingly, libraries focus mainly on infrastructure and databases/repositories as they are among the main focuses of the library sector in Hungary [10].

Significant progress was made between 2018 and 2020, when the Hungarian Electronic Information Service National Program (EISZ) concluded transformative open access agreements with major publishers [11]. As a result, communication related to open science also shifted to the promotion of gold open access publishing. At the same time, the broader interpretation of open science appeared first through international conferences such as Focus on Open Science, Budapest series [12], and later an increasing number of local events were dedicated to this topic as well.

In the period between 2019 and 2022, the open science focus on the strategic level shifted towards open and FAIR data management in Hungary. It seems as though we are at the beginning of a similar journey, on which we have already made significant progress in relation to green open access. The Hungarian research community focuses on the infrastructure, building the first data repositories, analyzing the arguments, creating strategies and trying to reach out and engage the researchers [13].

We can state that open science is receiving increasing attention from researchers, information specialists, strategy- and decision-makers and the public in Hungary. The National Position Paper on Open Science was signed and developed by the National Research, Development and Innovation Office (NKFIH) and other research related institutions and organizations in 2021. As we see, during these two decades, several papers discussed the current position and the benefits of open science on a strategic level as well as the difficulties of engaging the researchers in practice. Still, we can find few studies on best practices of promoting open science in the Hungarian research environment. These papers [14,15] support our assumption that researchers' achievement and commitment to open science is only possible with well-planned and well-prepared communication, where the practical implementation with the local environment and for the local researcher community must be emphasized in order to achieve results.

We analyzed the audience diversity of four online Hungarian Open Science Forum events organized as part of the NI4OS-Europe project. The anonymized dataset based on registration forms was filtered by profession. We attempted to identify the main differences of events attracting mostly librarians and those where the majority of the audience were researchers.

An open science newsfeed was also introduced by KIFÜ as part of NI4OS-Europe. The growing interest toward this newsfeed gave us a chance to analyze usage data of the 133 most-visited posts published in the period between May 2021 and April 2022, with the number of visitors until 30 June 2022. This gives us the opportunity to learn about the characteristics of the most visited posts. Data on the newsfeed were collected from KIFÜ's newsfeed log using Matomo Analytics. The newsfeed was analyzed through simple descriptive statistics, and the possible significant differences among variables were examined to validate hypotheses that the views of the posts depend on the selected criteria. The possible significant differences among variables were examined through the independent *T*-test. Where the sample sizes and variances were unequal between the groups (Leven's test $p < 0.05$), we used Welsch's *t*-test. The statistical analyses were carried out by using the software SPSS 21.0. For deeper analysis, a post classification was processed along the following groups: open science in general, open access, open and FAIR data, open methods, citizen science, open science infrastructure and financing open science (see Figure 1). Due to the significant overlap amongst the open science fields in the content of newsfeed items, they cannot be used as independent variables. In addition, group size after distribution is too small in many cases. Therefore, the comparison of usage data for these groups using mathematical statistical methods cannot provide reliable results.

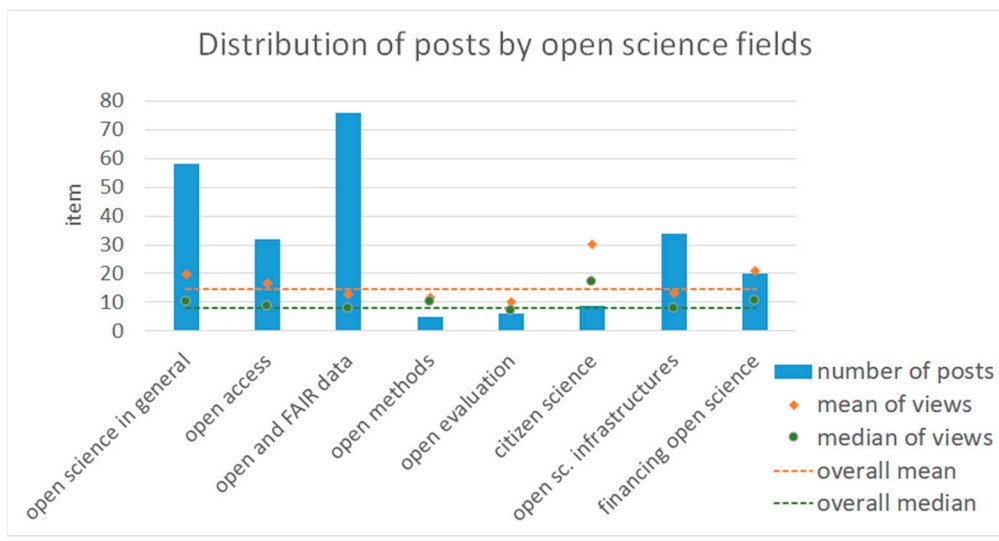

**Figure 1.** Distribution of posts by open science fields.

Due to the very small sample size, all the results of these studies need to be handled cautiously. Possible bias both in online traffic and event participations cannot be entirely precluded.

## 3. Results

As described above, one of the main objectives of the NI4OS-Europe project is to promote open science in Central and East European countries. The original concept counted on real-life activities, such as seminars and other on-site events. The rapidly spreading COVID pandemic and the restrictions it has caused made it impossible to conduct on-site events in 2021 and H1 2022, and all promoting efforts had to focus on online activities. This caused major changes compared to the original plans; however, being online only might have helped to reach out to a wider audience. Having all activities online also made it easier to archive all the materials and lectures and made them available for any possible later re-use.

All the promotion activities carried out as part of the NI4OS-Europe project were aimed at different, sometimes overlapping groups in Hungary, e.g.,

- publishing researcher interviews on research data management and open science practices for early career researchers;[1]
- open science news feed for open science practitioners;[2]
- publishing an e-learning course on EOSC and open science for graduate and PhD students;[3]
- testing RDM tools for the Hungarian research community;
- organizing various events, including the Hungarian Open Science Forum targeting senior researchers and stakeholders;
- an EOSC Champion program at three major Hungarian universities.

### 3.1. Researcher Interviews

The idea of publishing interviews with prominent researchers is considered as a bottom-up approach. Having no pressure or expectations, researchers could freely talk about their experience and practices regarding research data management (RDM), also giving voice to their concerns regarding open science. Interviewees were chosen from the widest possible range of research areas to show that RDM cannot be narrowed only to science, technology, engineering and mathematics (STEM) fields. All the interviews were published on Videotorium, which is the main Hungarian online video sharing platform of research and educational videos run by KIFÜ. A channel 'EOSC and Open Science' was launched to accommodate all the interviews, freely available to all.

Altogether, seven videos were published, and 396 views were recorded until 30 June 2022 (see Table 1).

**Table 1.** Video interviews published as part of the Hungarian NI4OS-Europe activity until 30 June 2022.

| Interviewee | ORCID | Research Field | Interview Publication Date | Number of Views until 30 June 2022 |
|---|---|---|---|---|
| Zoltán Kmetty | 0000-0002-6775-8938 | sociology | 17 March 2022 | 94 |
| Gábor Palkó | 0000-0002-4394-8577 | literary history | 17 June 2021 | 80 |
| Tamás Ferenci | 0000-0001-6791-3080 | biostatistics | 17 June 2021 | 79 |
| András Perczel | 0000-0003-1252-6416 | biochemistry | 16 June 2021 | 70 |
| Zoltán Kis and László Szentmiklósi (joint interview) | 0000-0002-8365-8507; 0000-0001-7747-8545 | physics and chemical engineering | 16 March 2022 | 34 |
| Miriam Szőcs | NA | art history | 17 March 2022 | 26 |
| György Eigner | 0000-0001-8038-2210 | system engineering | 27 September 2021 | 13 |

It might be tempting to group views according to research fields. However, the relatively small sample size, both of videos and views, prevent us from rushing to any conclusion. It might be worth considering whether alternative video sharing platforms in addition to Videotorium could help generate more views, especially noting that YouTube has become one of the largest search engines in the world [16].

*3.2. Open Science Newsfeed*

The open science newsfeed of KIFÜ was launched for the test phase in April 2021, and the live, daily–weekly updates were started later[4]. According to KIFÜ's role in open science in Hungary, this online newsfeed is an important source of information on open science for the Hungarian community. Therefore, analyzing the KIFÜ's newsfeed might show us what kind of open science information the Hungarian open science representatives are trying to convey to the community. On the other hand, we can see how this attempt meets the audience's interest.

The newsfeed informed researchers and stakeholders about international and local open science trends and events. Not all the posts were related directly to NI4OS-Europe; rather, the aim was the widest possible range of information. Most of the events promoted via the newsfeed were organized by different European associations or institutions. The main focus was on NI4OS-Europe and EOSC-related news, while many posts called the attention to press releases, policy papers, research articles and other publications regarding all aspects of open science.

The analysis covers the 133 most-visited posts that were published in the open science newsfeed of KIFÜ between 5 May 2021 and 7 April 2022. Online traffic until 30 June 2022 was recorded by Matomo Analytics. We collected the number of individual views of each item, filtering out multiple viewings by the same users. Then, we categorized the posts according to different aspects such as type, topic focus (international or local focus) and open science field. During the analysis, the distribution of items by category can be considered as a representation of the communication goals, while the number of views is a representation of the needs of the target group. Considering the shortcoming that the target group could only choose from what was presented in the newsfeed, analyses were carried out through simple descriptive statistics and examining possible significant differences among variables [17] (Falus and Ollé, 2008) by using the software SPSS 21.0.

For the first analysis, posts were grouped into two categories: 'event' and 'other'. Posts advertising some event (invitation to a future event or report of a past event) were categorized as 'event'. 'Other' posts discussed press releases, open science trends, EOSC news, reports, etc. Data show that 81 posts (61% of all posts analyzed) advertised an event, while 52 posts (39%) were categorized as 'other'. On one hand, this rate is a facility, and on the other hand, it can be a conscious communication strategy to achieve a more lasting impact and a greater commitment through an event. On the other hand, usage statistics show a slightly different ratio: while events gained 71% of total page views, other posts gained 29% only (see Figure 2). Carrying out an independent-samples T–test, we found that the mean of views for event and non-event type ($p = 0.07$, t = $-1.826$, F-test $p = 0.083$) are not significantly different.

Second analysis also grouped posts into two categories based on 'internationality'. Posts related to reports, press releases, events, etc. of institutions outside Hungary were labeled as 'international'. Events organized by Hungarian institutions, and posts related to Hungarian trends were labeled as 'Hungary'. The internationality of the items in the newsfeed shows that 98 posts (74%) were international, and only 35 of them (26%) presented local content. This means that the majority of information on open science is still coming from abroad. However, if we compare the views of the posts in distribution of international and local content, we see an opposite result: the local content was visited more often (See Figure 3). The independent-samples *T*-test with Welsch's test also validates that the mean number of views for international posts and for local posts is significantly different (Welsch's test: $p < 0.001$, t = 4.050; F-test $p < 0.001$). The average views for local posts were 20.473 more than the average views for international posts. This result shows that despite the fact that more international posts are available, the target group is more interested in local information.

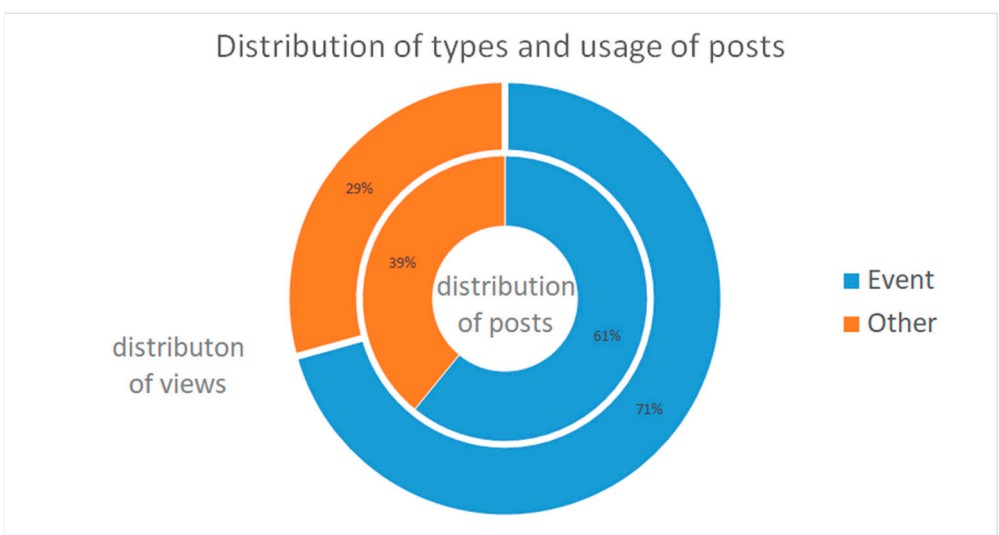

**Figure 2.** Distribution of types and usage of posts.

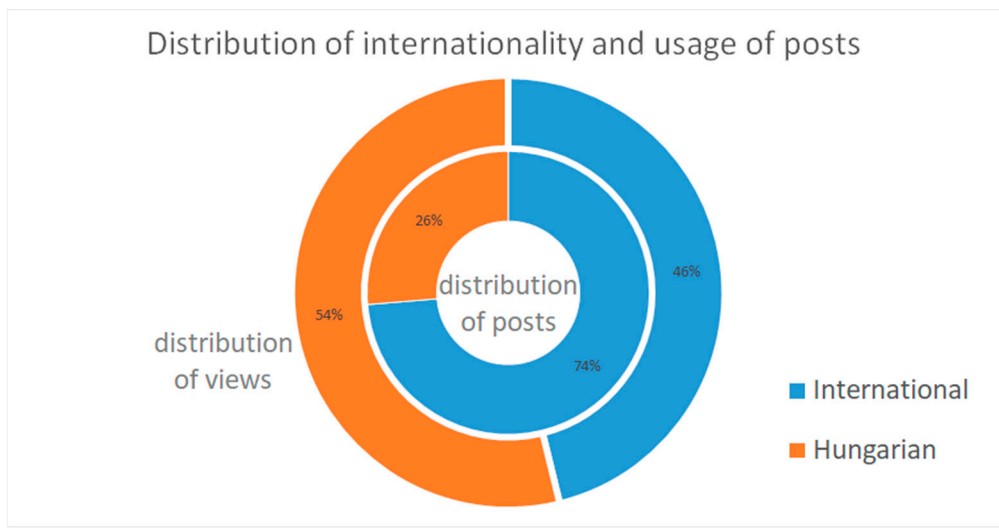

**Figure 3.** Distribution of internationality and usage of posts.

Since the events had greater presence in the posts, we decided to analyze it in more detail. The distribution of the 81 events related posts by internationality shows a very similar picture to what we saw when examining the internationality of the entire sample (see Figure 4). The presence of international events was much higher amongst the posts: 58 (72%) were labeled as 'international', while 23 of them (28%) were Hungarian. The usage statistics of posts was, however, lower compared to the local events: international events gained only 44%, while Hungarian events gained 56% of all page views. The Welsch's test validates that the mean number of views for posts on international events and for posts on local events is significantly different (Welsch's test: $p = 0.003$, t = 3.323; F-test $p < 0.001$). The average views for posts on local events was 22.629 more than the average views for posts on international events. This result shows that despite the fact that more international events are available on open science, the target group is more interested in local events. This is particularly an interesting result in light of the fact that, due to the pandemic situation, events were mostly organized online, so participation in international open science events did not involve more efforts than participation in local events. Various reasons can stay behind this phenomenon. One of these factors could be the language barrier, when a librarian or a researcher feels no ease at joining an English language event. While language barrier can be easily one of the main drivers of the unequal distribution

of local and international event post visitor numbers (see Figure 4), this clearly cannot be the only reason why posts on local trends gain more visitors than international ones (see Figure 3).

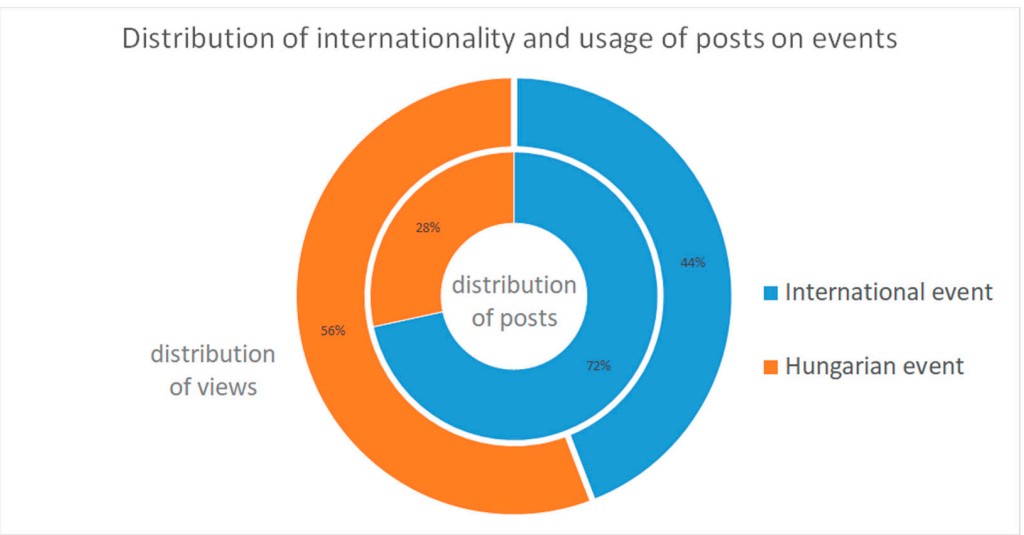

**Figure 4.** Distribution of internationality and usage of posts on events.

Grouping the newsfeed items by open science fields adequately shows the recent focus of the open science representatives in Hungary (see Figure 1). The hottest topic during this period was the open and FAIR data, as they were related to 57% percent of the newsfeed items, while the second and third specific fields (open science infrastructures and open access) were related to 25% and 24% only. Looking to the end of the list, citizen science (7%), open evaluation (5%) and open methods (4%) seem less popular or less important in the communication and open science promotion in Hungary.

Using descriptive statistics to analyze the covered open science fields, we see that the means of views as well as the medians of views are quite close to the overall, except for field citizen science (see Figure 1). This means that the interest of the newsfeed users did not differ from the received content distribution. The only exception was the citizen-science-related posts where users showed more activity. We can conclude that the focus of communication is on open and FAIR data management, while other areas receive less attention. This result can help in shaping the open science communication strategy in Hungary.

### 3.3. E-Learning Course on EOSC

Another open science promotion activity was the development of a Moodle-based e-learning course 'Open Science and EOSC in practice' by the KIFÜ team on H1 2021. The course was launched early July 2021, while an online workshop promoting the course and discussing possible developments took place on 13 July 2021. The course began with four modules, while an additional fifth module was added in January 2022. As the main target was student groups, the courses were launched in Hungarian only, and no prior knowledge was required. Modules are richly illustrated with short videos, diagrams, quizzes, and other interactive tools to make it easy to integrate them with any higher education course.

The five modules of the course can be handled individually, discussing different aspects of EOSC and open science (see Table 2).

**Table 2.** Modules of the e-learning course 'Open Science and EOSC in practice'.

| Module | Title | Key Topics |
|---|---|---|
| Module 1 | A practical guide on how open science can support researchers | Replication crisis, OS tools, OS community building, OS requirements of Horizon Europe calls |
| Module 2 | FAIR data management | FAIR principles, DMP tools, financial aspects of data management |
| Module 3 | Changes driven by EOSC | EOSC development phases, researcher contribution to EOSC, EOSC working groups |
| Module 4 | EOSC services | EOSC portal and marketplace, EOSC on-boarding process |
| Module 5 | Trends of the research systems | OS metrics, research evaluation |

We collected feedback on the testing of the educational module developed for teaching open science, for which we used a workshop as the most effective method. This direct communication ensured the correct and accurate interpretation of the information received from the testers. The modules were tested in advance by 40 people, mostly librarians, of 15 research and higher education institutions. Results of the tests and views of the people involved in the testing were discussed during the workshop organized on 13 July 2021. The feedback praised the interactivity of the course and the rich illustration materials. One attendee also added that 'it would be awesome if science could work in this [Open Science] way'.

It also became clear during the workshop that integrating the course with the university curricula had certain barriers. Open science usually is not considered as individual courses, and some of the Hungarian universities lack even obligatory general courses regarding research support or research methodology. Open science and research methodology are considered often as 'library businesses', while librarians have no full-semester courses and are invited to contribute existing courses only occasionally.

Based on these responses, the e-learning course was opened and recommended for the EOSC Champion program to gain more feedback.

*3.4. EOSC Champion Program*

The EOSC Champion program was devised as a series of nine events at major Hungarian universities. While mentoring is considered to be more effective in a multi-year connection [18], these one-year-long champion programs might also have a positive effect on the early career researchers' career path by showing an alternative research methodology.

To run this program, cooperation was built between KIFÜ and university professors who promoted EOSC and open science among their fellow researchers and PhD students. Altogether, three universities took part in the program: Eötvös Loránd University, Óbuda University and University of Szeged.

About one third of the program focused on research data management issues, while two events specifically discussed EOSC as follows:

1. Introduction
2. Open Science
3. Scientific Publications
4. Research Data Life Cycle and Management
5. Data Management Plan
6. FAIR Principles
7. EOSC—overview
8. EOSC—services
9. Future Trends in Scientific Research Systems and Careers

These were held as a monthly series, where KIFÜ provided help for the professors for each event, including PowerPoint slides, a list of questions possible to drive open discussions and further publications on the recent development of the certain topic. Events were held mostly as in-person seminars, occasionally changed to online. Monthly project meetings were also conducted for the champions to share experiences and discuss oncoming topics.

Though systematic surveying of the audience did not take place, all three EOSC champions shared their thoughts and feelings during the monthly meetings about this program. The overall conclusion of the EOSC Champion program was that PhD students are more likely to take part in the open science discussion (even though this series was not officially part of the PhD curriculum), while professors are much harder to involve in such activity. Taking part in any kind of open science activities is barely acknowledged in the research evaluation process of the universities, and this is not easily overcome for individual researchers. While most of the attendees agreed on the global benefit of open science, much feedback was gained about the lack of the financial drivers regarding open research practices.

### 3.5. Testing RDM Tools

The issue of data repositories has found its way to the hot topics of research policy in recent years in Hungary, and most of the universities and research institutions still provide no such service for their researchers. A cooperation was formed between KIFÜ and the National Laboratory for Digital Heritage (DH-LAB) that serves as a major center for digital humanities in Hungary. The DH-LAB was officially founded in late 2020, while earlier works were also carried out at the same institution [19]. With support from DH-LAB expertise, Invenio was chosen to build repository structure that is freely adaptable to institutional requirements, while technical support is provided by KIFÜ. Invenio is still in beta, which makes it possible to easily shape the required features of the future repository, while being a CERN software makes it safe enough to build on. This work started in H1 2022, and results will be published only at the end of the project. Parallel with and independent of this initiative, other data repository projects commenced in 2022, most importantly the Eötvös Loránd Research Network (ELKH) ARP Data Repository project[5] that aims to provide a data repository covering researchers of the largest publicly funded Hungarian research network of 11 research centers, 7 research institutes and 116 additional supported research groups.

These initiatives are still under development, so we have no possibility to judge any of the outcomes. The many independently started RDM projects, however, make clear the awareness of research data, FAIR principles, and open science criteria both from researchers and research funds sides.

### 3.6. Hungarian Open Science Forum

The Hungarian Open Science Forum was launched as part of the NI4OS-Europe project by the two Hungarian consortium partners: DEENK and KIFÜ. The forum is organized as an online event. Skiles et al. [20] discusses all the benefits of online events, including costs and wider geographical composition of attendees. These factors were clearly a benefit of the online format, while organizers had to be more conductive to generate discussion.

The main objective is to inform the Hungarian researcher community about recent open science, especially EOSC-related trends. Another important aspect of the forum was to introduce and gain feedback regarding the then-forming National Position Paper on Open Science.

A forum event is usually 90–120 min long, where some presentations are held, and online discussion is formed based on the presentations. For H1 2022, four forum events were organized: The first forum took place on 28 May 2021, followed by the next ones on 24 September 2021; 19 January 2022 and 28 April 2022. Topics of the forum events varied from discussing open science policies of European countries, introducing the National

Position Paper on Open Science and presenting open science practices for life scientists and social scientists (see Table 3).

**Table 3.** Number of attendees of the Hungarian Open Science Forum events.

| Event | Main Topic | Date | Number of Attendees |
|---|---|---|---|
| Hungarian Open Science Forum I | Introducing open science and EOSC | 28 May 2021 | 53 |
| Hungarian Open Science Forum II | Open science policies of European countries | 24 September 2021 | 36 |
| Hungarian Open Science Forum III | National Position Paper on Open Science | 19 January 2022 | 138 |
| Hungarian Open Science Forum IV | Open science practices for life scientists and social scientists | 28 April 2022 | 74 |

All the Hungarian research institutions and higher education institutions were informed about the forum events directly via e-mail. A total of 150–270 e-mails were sent to promote each event, while social media communication also supported the recruitment process. The number of attendees varied between 36 and 138 (see Table 3). For the analysis, all data of registrants who had not attended the meeting were removed from the dataset.

The third forum had the highest number of attendees. This event was organized together with the National Research, Development and Innovation Office, that is the main research fund body of Hungary. At this event, the vice president for science and international affairs and the open science advisor of the office introduced the newly launched National Position Paper on Open Science. It seems clear that the National Research, Development and Innovation Office attracted more attendees than other forums.

Due to the registration form, we were able to analyze the attendee affiliations and professions (see Table 4). For this, three groups were formed: researchers, librarians (meaning all library staff, including IT specialists and data stewards) and organizers (all KIFÜ and DEENK staff). Where the 'profession' field was left blank during registration, affiliation and e-mail fields helped to determine the most suitable group for the attendee type.

**Table 4.** Number of attendees of the Hungarian Open Science Forum events broken down by affiliation.

| Event | Number of Attendees | Number of Researchers (Among Attendees) | Number of Librarians (Among Attendees) | Number of Organizers (Among Attendees) |
|---|---|---|---|---|
| Hungarian Open Science Forum I | 53 | 23 | 24 | 6 |
| Hungarian Open Science Forum II | 36 | 7 | 19 | 10 |
| Hungarian Open Science Forum III | 138 | 82 | 45 | 11 |
| Hungarian Open Science Forum IV | 74 | 19 | 43 | 12 |

The forums are generally visited by more librarians than researchers. The only exception is the third forum where the National Research, Development and Innovation Office as co-organizer might have attracted researchers' attention. This effect, however, did not occur at other events. This suggests that open science is indeed considered as a 'library business'. This was reflected by heads of research, who answered the invitation letters, and delegated the librarian of the institution to the forum. The high ratio of librarian attendees of the

events also underlines the importance of librarians in promoting open science. Librarians play a large role in facilitating open science in their research institutions.

## 4. Discussion

This paper describes all the open science promotion activity carried out in Hungary by KIFÜ as part of the NI4OS-Europe project. Several activities were introduced, including online interviews with researchers, champion programs for early career researchers, an e-learning course, open science newsfeed, online events, etc. By learning the usage data and attendee ratio of 2021 and H1 2022 activities, we might have identified practices that proved to be more successful in Hungarian context compared to others. These data need to be handled cautiously due to small sample size; results are more likely only impressions of the first half of the project activities.

Studying usage and number of visitors, online video interviews and e-learning courses were less attractive in the period, noting that these activities are not meant to be used one-time only. It is clear, however, that additional promotion is needed to reach the target audience, especially via social media and YouTube. The overall impression of an EOSC Champion program was that younger researchers are easier to involve in open science discussions. The greatest barrier seems to be that open science activities are barely acknowledged in the research evaluation process of the researchers.

Relatively high usage and a wide range of posts were recorded for the Hungarian NI4OS open science newsfeed. Analyzing the posts from different aspects we saw that events were overrepresented, and the feed had much more international than local context. The usage of posts confirmed the strategy of overweighting the events as the interest for these posts was greater than expected. The usage data of the posts revealed that although more international posts were available to the users, they still read the ones with local context more. The same results can be seen by analyzing the usage of posts for international and local events. Despite the fact that, due to the pandemic situation, the international events were also held online, the posts of domestic events received much more interest. This result may even indicate language barriers.

By examining which open science areas the posts focused on, we found that the topic of open and FAIR data management was given priority in the communication. The usage of the posts in relation of open science fields showed balanced attention from readers, confirming that the content provided in the newsfeed met the needs of the community. However, it is also important to take into account that the offered content influences the consumption of information, so in open science communication it may be worthwhile to give space to fields that are currently receiving less emphasis.

Analyzing the attendee ratio of four online Hungarian Open Science Forum events, it seems that open science is considered part of a librarian's duty, while researchers can be involved in higher numbers when research funds are also a factor. This suggests that bottom-up open science promotion activity needs to be accompanied by a top-down approach as well. Based on the results, the role of librarians is particularly important in facilitating open science, so emphasis must be placed on their training in this direction to provide the appropriate skills for knowledge transfer.

This study might be of great help in mapping the open science landscape of a Central European country, being the first to assess the promotion activity of an open science project in Hungary. Using multiple data sources for this purpose, we have the possibility to form conclusions of various aspects of the Hungarian open science landscape. This is one of the first data-based research studies that can point out that open science is clearly linked to libraries, and it is generally thought that open science is role for librarians. This finding might help shape the skills of library and information professionals in the future. The paper also shows that the language barrier can be measured in a Hungarian context. This underlines the importance of using national language(s) when promoting open science and science in general. While using English is a must when following international trends,

initiatives of organizing local events and translating statements, white papers cannot be underestimated.

To learn more about the results, it would be worthwhile to collect data from other NI4OS-Europe consortium members regarding their open science promotion activities. This would make it possible to compare usage patterns of different Central and South European countries and learn if the above results can be observed for other research communities.

**Author Contributions:** Conceptualization, Á.L.; methodology, Á.L. and P.S.; formal analysis, P.S.; investigation, Á.L. and P.S.; resources, Á.L. and P.S.; data curation, P.S.; writing—original draft preparation, Á.L.; writing—review and editing, P.S.; visualization, P.S.; supervision, Á.L. All authors have read and agreed to the published version of the manuscript.

**Funding:** This research was funded by the European Commission under the Horizon 2020 European research infrastructures grant agreement no. 857645.

**Data Availability Statement:** Research data regarding this study are openly available via Zenodo. Lencsés, Á.; Sütő, P. Number of Attendees of Hungarian Open Science Forum Events, and Number of Visitors of KIFÜ Open Science Newsfeed, 2022. https://doi.org/10.5281/zenodo.7034816 (accessed on 30 August 2022).

**Acknowledgments:** We thank Krisztián Kovács (KIFÜ) for assistance with using Matomo Analytics.

**Conflicts of Interest:** The authors declare no conflict of interest.

## Notes

1.    The interviews are available (in Hungarian only) at https://videotorium.hu/hu/channels/4935 (accessed on 14 October 2022).
2.    The newsfeed originally was published under https://kifu.gov.hu/ni4os/hirek. During Summer 2022, the major redesigning of KIFÜ's web-page concluded in a new site under https://kifu.gov.hu/ni4os-hirek/. All earlier posts have been migrated to the new platform, and earlier URLs have been redirected to the new sites (accessed on 14 October 2022).
3.    Nyílt tudomány és EOSC a gyakorlatban. Available online: https://elearning.kifu.hu/ (accessed on 14 October 2022).
4.    For the newsfeed availability, see note 3.
5.    The acronym ARP refers to 'data repository project' in Hungarian: AdatRepozitórium Projekt.

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
