# Peer review of "Challenges of Promoting Open Science within the NI4OS-Europe Project in Hungary"

_publications, doi:10.3390/publications10040051_

Round 1
Reviewer 1 Report
The manuscript is solid and very well written it describes all the open science promotion activities carried out in Hungary. The promotion activities could be used to transmit knowledge to a significant number of stakeholders (librarians, funders, researchers) in an effective and efficient way.
Very well referenced. The aspect of promoting Open Science It is increasingly recognized as a crucial factor in bridging the science, technology and fulfilling the human right to science the manuscript is clearly focused.
Reviewer 2 Report
At the beginning, each promotional activity that was carried out to promote open science should be specified through a graphic representation, and in a timeline. On the other hand, it is necessary to add the URL where the content of the programs, the links to the videos of the interviews, and the mentioned activities can be reviewed. The scientific method used for each of the activities should be specified, as well as a description of the good practices used to evaluate the contents, that is to say, the validity and reliability of the instruments used; in an orderly manner, indicate the sample used for each of the activities in its respective section. If possible, add these references:
González-Pérez, L. I., Ramírez Montoya, M. S., & García-Peñalvo, F. J. (2022). Technological enablers 4.0 to boost open education: input for UNESCO recommendations. RIED-Revista Iberoamericana De Educación a Distancia, 25(2), 23-48. https://doi.org/10.5944/ried.25.2.33088.
Ramírez-Montoya, M. S. (2020). MOOCs and Developments and OER:
contributions for open education and open science. In Radical Solutions and Open Science (pp. 159-175). Springer. https://doi.org/10.1007/978-981-15-4276-3
Reviewer 3 Report
The paper aims to highlight challenges of promoting open science within the NI4OS project. The paper describes nicely the history from green open access to gold open access and recognises similar developments towards open data and fair, but with a stronger focus on infrastructure requirements and research culture changes that need to be addressed in support of open science.
The analysis is rather narrow and based on NI4OS-Europe events and KIFÜ newsfeeds and the authors mention this. The authors also mention a bias likely introduced by event participants visiting the event news feeds. It is not mentioned in the paper how many news posts were actually related to event posts and how many events actually happened (in person/on site). The paragraph describing this (lines 100 to 117) is not easy to understand and should be improved. The results later indicate that the analysis revealed that 61% of posts were related to events. The analysis is mixed with the results and this is maybe why the paper is really difficult to read.
While figure 1 shows percentage, the text talks about numbers. The statement is not complete and figure 1 and 2 not clear. The authors must have missed re-reading the paragraph "If we compare the views of the two types of posts [event and other], we can see a higher rate of interest at the views metric of the event type than the number of items 184 metric of the event type (see figure 1)."
How is "internationality" defined? Does this mean that the event was held outside Hungary? Or that the newsfeed was not about an event but about a topic written by an author outside Hungary? Since 'internationaliy' takes up quite a substantial part of the entire paper the authors might want to define it.
The grouping of the posts (figure 4) would have been much more supportive for the reader had they been posted in the beginning of the paper. They are describing the data source, not a result. Figure 4 has two different number scales at each side. Why and how should these be understood?
It is not entirely clear as to how 3.3 contributes to the actual analysis. If this was meant to highlight barriers - the barriers should be indicated or the challenges outlined as per the results from the testers or institutions.
The paper needs substantial revision.
The underlying datasets have been made available and are mentioned and meet funder requirements in that a data availability statement at the end of the paper is provided. A reference to free video recordings is also provided and there might be options to post videos to YouTube and expand the study later. It might be worthwhile to visit http://videolectures.net/ for future studies.
Round 2
Reviewer 2 Report
You still need to indicate the method you used and the analysis technique for each representation. It is not yet clear. You should add in the conclusions section the new knowledge that emerges from this analysis.
Author Response
Thank you again for the review and your comments. Methodology has been enriched, and is now part of all chapters where any analysis took place. Results and conclusions are also explained in more detail. The ‘Discussion’ chapter of the manuscript has been also extended with results of the research.
Reviewer 3 Report
The authors clearly addressed the highlighted issues. The paper now reads well, and the results are nicely presented. The study/methods and insights are really interesting despite the small sample size.
Author Response
Thank you again for the review. We do think that your comments and suggestions helped to shape the manuscript to a more clear and easy-to-follow article.